DOI: 10.1038/s41467-018-07562-7　　**OPEN**

# Synchronization of speed, sound and iridescent color in a hummingbird aerial courtship dive

Benedict G. Hogan [1,2] & Mary Caswell Stoddard [1,2]

Many animal signals are complex, often combining multimodal components with dynamic motion. To understand the function and evolution of these displays, it is vital to appreciate their spatiotemporal organization. Male broad-tailed hummingbirds (*Selasphorus platycercus*) perform dramatic U-shaped courtship dives over females, appearing to combine rapid movement and dive-specific mechanical noises with visual signals from their iridescent gorgets. To understand how motion, sound and color interact in these spectacular displays, we obtained video and audio recordings of dives performed by wild hummingbirds. We then applied a multi-angle imaging technique to estimate how a female would perceive the male's iridescent gorget throughout the dive. We show that the key physical, acoustic and visual aspects of the dive are remarkably synchronized—all occurring within 300 milliseconds. Our results highlight the critical importance of accounting for motion and orientation when investigating animal displays: speed and trajectory affect how multisensory signals are produced and perceived.

[1] Department of Ecology and Evolutionary Biology, Princeton University, Princeton, NJ 08544, USA. [2] Rocky Mountain Biological Laboratory, Crested Butte, CO 81224, USA. Correspondence and requests for materials should be addressed to M.C.S. (email: mstoddard@princeton.edu)

Some of the most extraordinary and elaborate signals in nature are used by animals in courtship displays[1,2]. Not only are these displays often multimodal, stimulating multiple senses, they typically include motion: peacocks (*Pavo cristatus*) rattle their trains[3,4], jacky dragons (*Amphibolurus muricatus*) flick their tails[5], and spiders (*Habronattus* spp., *Schizocosa* spp.) dance, producing vibratory signals in the process[6–9]. Investigating complex and dynamic displays—especially in the wild, and from the perspective of the signal receiver—has proven to be a major challenge, technically and conceptually[10,11]. As a result, we lack a fundamental understanding of how complex displays work[12,13]. How does the dynamic nature of a display (motion, speed) influence signal transmission? How synchronized are the different components of a multimodal display?

A new framework for analyzing complex signals—rooted in systems theory—has recently been proposed[12,14]. This framework emphasizes the importance of understanding the relationship between structure and function in signal design, particularly in the context of signals that are multimodal (involving more than one sensory modality) and dynamic (variable in time and space). It also highlights the critical role of the signal receiver, whose sensory and cognitive processes exert strong influence on signal design[15,16]. Multimodal signals are common in courtship behaviors[11,17–20] and can have diverse functions[13,21] related to their informational, aesthetic, and sensory content[22]. For example, multimodal signals might convey multiple messages that reveal different information about male quality[13], or they might reflect female aesthetic preferences[23]. From a sensory perspective, multimodal signals may have evolved because they are efficiently transmitted, detected or remembered[13]. The different signal components might amplify or reinforce one another, provide backup in variable conditions (e.g., a song still works on a foggy day, when visibility is low), or may operate at different distances[13]. In many cases, multimodal signals are also dynamic, and courtship displays in particular often have crucial motion-based elements[1,24]. The vigorous strut display of sage grouse (*Centrocercus urophasianus*)[25] and the coordinated dances of lance-tailed manakins (*Chiroxiphia lanceolata*)[26] are classic cases.

Although work on multimodal signals is proliferating[12,14], we still know relatively little about the temporal organization of complex displays—and the extent to which signal components are synchronized[2,11]. It is becoming increasingly clear, however, that the timing and ordering of signal components can have large impacts on signal function and perception. In particular, temporal organization is believed to play a role in sensory integration, which is the binding of sensory input from multiple senses (or multiple sources) into a coherent whole: integration can help animals correctly identify the shared source of separate unimodal signal components[18]. In humans[27,28] and macaques (*Macaca mulatta*)[29], for example, some degree of temporal synchronization is required for the integration of acoustic and visual aspects of conspecific vocalizations. Similarly, captive male poison-dart frogs (*Epipedobates femoralis*) will attack an artificial male conspecific only if its acoustic (synthetic advertisement call) and visual (vocal-sac pulsations) signals are synchronized (to within 434 ms)[30]. However, the assessment of synchronized signals is not always straightforward. Sister flycatcher taxa (*Monarcha castaneiventris* complex) differ in whether they assess song and color together or in sequence, even when the signals are presented simultaneously in a playback experiment[2]. And, remarkably, changing the order and timing of relatively unattractive display components (one acoustic, one visual) can perceptually rescue túngara frog (*Physalaemus pustulosus*) mating signals, creating a newly attractive display if played in the right combination[20]. Given this, understanding the production and perception of synchronized displays requires further study in diverse animal groups.

The 'bee' hummingbirds (tribe Mellisuginae), a colorful and rapidly diversifying clade[31], provide a powerful system in which to investigate complex, dynamic displays. Males of most species perform spectacular courtship dives at high speed over or next to the presumed signal receiver, usually a conspecific female[32–34]. Male broad-tailed hummingbirds (*Selasphorus platycercus*) typify these dives. Males display to stationary females by climbing vertically to ~30 m and making sequential powered (by active wing-flapping) U-shaped dives[32,35]. With their wings, males produce a non-facultative trill when in flight and when diving[36]. In addition, males generate a facultative buzz, only during dives, with the second retrix of the tail feathers[35]. Males also have a striking red iridescent gorget, which may have a role in signaling behavior[37,38]. Thus, broad-tailed hummingbird courtship dives appear to be complex[12,14], including multiple components and multiple modalities (acoustic, visual). The dives are also dynamic, with signal production—and probably perception—changing as a function of time and space (i.e., the male's position and orientation relative to the female). The degree to which different components of the male's dive display are synchronized is unknown.

We performed a comprehensive quantitative analysis, encompassing motion, sound and color, of hummingbird courtship dives. We quantified sound and color in a way that accounts for the perceptual experience of the presumed signal receiver (the female). In brief, we video recorded and analyzed 48 dives performed by male broad-tailed hummingbirds in the wild at the Rocky Mountain Biological Laboratory in Gothic, Colorado. We used image-tracking software to characterize the male's position and speed throughout the dive. We combined this with acoustic analysis to determine when in the dive the male produces mechanical noises (hereafter sonations). We then estimated what these would sound like to the female, accounting for the Doppler effect. Next, we used multi-angle ultraviolet-visible photography and a computational model of hummingbird color vision to determine how the female's perception of the male's iridescent gorget would change throughout the dive display. Finally, we determined the extent to which the key motion-based, acoustic and visual components of the dive are synchronized.

## Results

**Speed and sound.** During dives, males attain a maximum speed of ~23.25 m s$^{-1}$ (Fig. 1d; $n = 17$ video recordings containing 48 dives, SD = 5.82), which is comparable to that of Anna's hummingbird (*Calypte anna*) dives[33]. Dives were observed occurring in bouts of 2–8, with each dive (descent plus ascent for a subsequent dive) lasting about 6.4 s ($n = 18$ audio recordings containing 72 dives, 24 of which were excluded from video analysis, SD = 0.72). Audio analysis revealed that the wing- and tail-generated sonations produced during broad-tailed hummingbird courtship dives are highly stereotyped and can be separated into three general sections that always occur in a precise order (Fig. 1a, b, Supplementary Fig. 7). The sonations were assigned as wing- or tail- generated based on previous descriptions of these sounds[35,36]. Section A (Fig. 1a, b, purple) corresponds to wing-generated sounds emitted during the downward and horizontal parts of the dive, which are punctuated by a well-described mechanical buzz generated by the tail feathers[35]. Sections B and C are characterized by variable numbers of short (Fig. 1a, b, teal) and then long (Fig. 1a, b, pink) wing-generated sonations, which correspond to climbing flight as the hummingbird ascends to start a subsequent dive. These acoustic components are dynamic: due to the high speeds involved, we estimated that the female's perception of the acoustic frequencies of the (wing- and tail-generated) sonations shifts up by ~6.5% (as the male approaches,

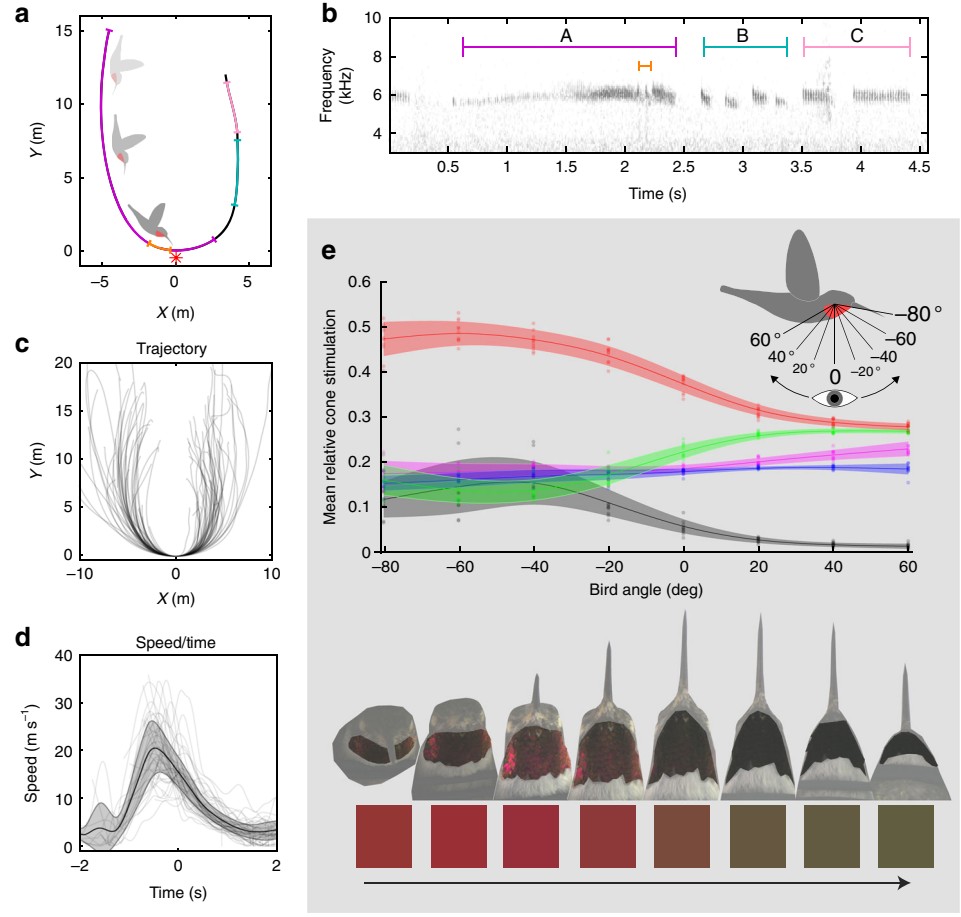

**Fig. 1** During the dive, sound and color change markedly as a function of trajectory and speed. **a** Representative dive aligned with the origin at the nadir (point of lowest height). The asterisk indicates estimated female position. Colored sections correspond to the sonogram of sonations during the dive. **b** Sonogram corresponding to the representative dive. Colors indicate sections of dive sonation. Purple: wing-generated main dive sonations (section A). Orange: tail-generated sonations. Teal: wing-generated short sonations (section B). Pink: wing-generated long sonations (section C). **c** Plot of all tracked dives aligned as above and overlaid ($n = 48$). **d** Mean estimated speed during the dive. Shaded area indicates standard deviation ($n = 17$ bouts of diving containing 48 dives). Also overlaid with transparency are individual measures from all tracked dives ($n = 48$). **e** The perceived color of the gorget changes from red to black. Above: Female's relative cone stimulation values, estimating perception of the male's gorget (based on 10 male specimens), as predicted by a hummingbird vision model as a function of male orientation. Negative x values indicate that the beak is rotated toward the observer. Positive x values indicate that the beak is rotated away. Points indicate measured values, lines show smoothing spline interpolation of the mean, and the shaded area indicates standard deviation of the mean. Red: longwave-sensitive cone. Green: mediumwave-sensitive cone. Blue: shortwave-sensitive cone. Magenta: ultraviolet-sensitive cone. Black: double cone. Note that double cone stimulation is not relative to the other cones but to the reflectance standard in each image (0–1). Below: examples of extracted gorget images, and an illustration of the average hue of the gorgets at each orientation (irrespective of intensity/brightness). The arrow represents the direction of perceived color change that occurs for a female at the nadir of the dive; see main text below

mean estimate 6.53%, $n = 17$ video recordings, SD = 1.87) then down by ~4% (as he departs, mean estimate 3.98%, $n = 17$ video recordings, SD = 0.81) due to the Doppler effect (Fig. 2b).

**Iridescent color**. To determine how a female would perceive the male's iridescent gorget during the dive, we obtained ultraviolet- and (human) visible-photographs, taken from multiple angles, of male broad-tailed hummingbird museum specimens. Combining this information with tracked male position and assumptions about female position and male posture (see methods, Supplementary Figs. 2–5), we estimated that the gorget would only be visible to the female when the male's body is at certain angles. In an angular reference frame where 0° corresponds to the time at which the long body axis of the male is directly orthogonal to the female (Fig. 1e), his gorget would only be visible over the range of −80° (where the beak is pointed toward the female, as he approaches) to 60° (when the beak is pointed away from the

female, as he departs). Outside of these angles, the gorget is occluded by the rest of the male's body. We used a model of hummingbird color vision[39] to estimate how a female's photoreceptors would be stimulated by the gorget while it is visible as the male flies over her (Supplementary Table 1).

From the female's perspective, the male's gorget changes dramatically in appearance throughout the dive, shifting rapidly from bright red to dark green/black at the nadir (lowest point) of the dive (Figs. 1e and 2c).

This color change occurs very quickly, during the brief ~120 ms period of the dive for which the gorget is estimated to be visible to the female (mean 123 ms, $n = 17$ video recordings, SD = 17.5). When modeled in avian tetrahedral color space, the gorget colors sweep through the black-to-red regions (Supplementary Fig. 8): the perceived difference in color between views of −80° (when the gorget appears red) and 60° (when the gorget appears black, Fig. 1e) corresponds to ~14 just-noticeable

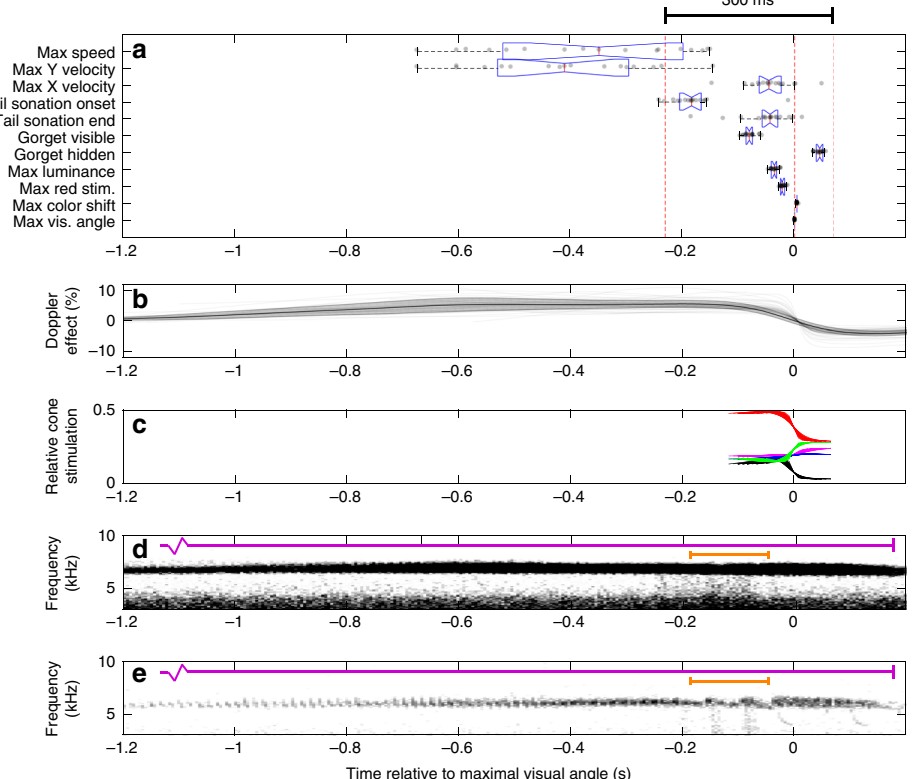

**Fig. 2** Horizontal velocity, sound and color are remarkably synchronized, occurring within 300 ms. **a** Box-and-whisker plot of measured and estimated times for various parts of the diving behavior: mean time of maximal estimated speed, mean time of maximal Y (vertical) and X (horizontal) velocity, mean times of tail-generated sonation onset and cessation, mean times of gorget visibility, mean time of maximal estimated luminance of the gorget, mean time of maximal estimated female LWS stimulation, mean time of maximal color shift, and the mean time of maximal female visual angle taken up by the male. N = 17 (17 bouts of diving containing 48 dives) for all measures, dotted lines indicate −230 ms, 0 s, and 70 ms. For each measure, box tails indicate 25th and 75th percentiles, and the central line indicates the median. **b** Mean estimated Doppler effect. Shaded area indicates one standard deviation (N = 17). Also plotted with transparency is estimated Doppler effect for individual dives. **c** Estimated female relative cone stimulation during the dive. Red: longwave-sensitive (LWS) cone stimulation. Green: mediumwave-sensitive cone stimulation. Blue: shortwave-sensitive cone stimulation. Magenta: ultraviolet-sensitive cone stimulation. Black: double cone stimulation. **d** Averaged sonograms of 30 dives, with minor contrast enhancement to highlight the darkening due to the low frequencies of tail-generated sonations. **e** Representative sonogram from one dive; colored bars correspond to sections in Fig. 1a and b. All values are temporally aligned such that the time of maximal visual angle, i.e., the nadir of the dive, is 0

differences (or a tetrahedral color span of 0.19, Supplementary Table 2). To capture the dynamic nature of this color change, we measured color shift—defined as the rate of change in the stimulation of the long-wavelength sensitive (LWS) cone during the dive (LWS was chosen because it represents the greatest magnitude color change with viewing angle, Fig. 1e). We also measured additional visual parameters, including the time during the dive at which the male takes up the largest portion of the female's visual field, as well as the time at which the gorget has greatest luminance (most stimulates the double cones[40]) and most stimulates the LWS cone.

**Synchronization**. The key physical, acoustic and visual aspects of the dive are remarkably synchronized (Fig. 2a). The male reaches top horizontal velocity, spreads his tail feathers to produce an audible buzz (Fig. 2d, e), and displays his gorget all within ~300 ms (46/48 dives < 300 ms; mean range 233 ms, SD = 24 ms, n = 17 video recordings)—roughly the duration of a (human) blink of the eye[41]. During this short time, the perceived frequency of the mechanical sounds shifts up (as the male approaches) and then down (as he departs, Fig. 2b), and the perceived color of the gorget shifts dramatically from bright red to black (Fig. 2c). An analysis of individual dives (n = 48 dives from

17 video recordings) shows that most dives follow this general temporal organization, with some minor variation (Supplementary Fig. 10).

## Discussion

Explaining the evolution of complex animal signals remains a major outstanding problem in biology[12,13]. Here, we addressed two critical but overlooked aspects of animal displays—temporal and spatial organization. Our results highlight the importance of accounting for motion and orientation (speed, trajectory) when investigating animal displays, since they exert strong influence on how multisensory signals (acoustic, visual) are produced and perceived. We found that sound and color in hummingbird dives are highly dynamic, changing dramatically in time and space. Due to the Doppler effect, we estimated that the female perceives a 6.5% shift upwards and 4% shift downwards in the frequency of the male's sonations as he approaches and departs, respectively. This change is directly related to the speed and trajectory of the male dive. In addition, the geometry of the dive—combined with high speed—dictates that the male's iridescent gorget is only very briefly visible to the female: during this time (~120 ms), the perceived color of the gorget changes rapidly from red to black.

What is the function of a complex, synchronized display? The broad-tailed hummingbird's acrobatic dive combines conspicuous acoustic and visual components, which are likely audible and visible, respectively, to females. These components are produced by the male—and likely perceived by the female—in a highly synchronized burst: the male maximizes horizontal velocity, mechanical sound and iridescent color display in a 300 ms period (within the range of a human blink[41]). The dive seems likely to be a multimodal signal, although we currently lack behavioral data—in this study specifically, and in hummingbird research generally—on female responses. Deciphering the function of multiple display traits in this system—in the same spirit as work on spiders[6–9], frogs[42,43], and warblers[44]—is a tantalizing (if challenging) next step. At present, we can only speculate about the function of the different multisensory components in the dive display[13,45]. The acoustic (wing- and tail-generated sonations) and visual (iridescent gorget) components might relay different information about the male's quality or condition, or together provide a mechanism for judging male flying skill[46]. Alternatively (or complementarily), the acoustic and visual components might provide backup in different environmental conditions (i.e., sound is more effective than color on an overcast day) or interact to increase detection or enhance memory[13]. Finally, the multimodal and highly synchronized dive of male hummingbirds might simply suggest "a taste for the beautiful"[47] and "the evolution of beauty"[48], resulting from strong selection by females' sensory biases or aesthetic preferences.

Synchronization may be a ubiquitous but understudied feature of complex animal signals[27–30,49]. Here, through a holistic analysis of a courtship display, we uncovered previously unquantified synchronization between visual and acoustic display components in hummingbird dives. In contrast to a proliferating theoretical literature on multimodal signaling in general[13], specific hypotheses for the function of synchronization are less well formulated[11]. In some cases, it appears that the proposed advantages of multimodal signaling[45] could extend to synchronization. For example, synchronization may allow two or more signals to provide new information in combination: perhaps accurately diving past a female at high speed and simultaneously producing mechanical sounds with the tail is difficult, and thus the degree of synchronization is informative about the male's flying ability[46]. Alternatively, synchronization of signal components may simply transmit the information in a shorter amount of time, thereby increasing the efficiency of the exchange. However, synchronization may have unique functions that are not related to those proposed for multimodal signals. For example, synchronization appears to be an important factor in the successful integration of sensory information from multiple modalities[18]. Thus, synchronization could be essential for effective neural processing of complex stimuli. A pipe dream for now, designing realistic robotic hummingbirds that could be programmed to dive over females while exhibiting varying degrees of synchronization of speed, sound and color could allow these alternatives to be explored—if we could determine a reliable way to assess female response. A more modest approach would be to capitalize on the existing variation in synchronization among males.

The tight synchronization of horizontal velocity, sound and iridescent color display (to ~300 ms) raises the question of whether the putative visual and acoustic signals in the hummingbird's dive are 'fixed' or 'fluid'—that is, whether they are produced synchronously due to physiological constraint (fixed) or whether their timing is independent of constraints (fluid)[50]. Because of the geometry of the U-shaped dive (Fig. 1c), the timing of gorget visibility and the accompanying color shift (along with any other visual components arising from the gorget) are likely to be fixed to the nadir. However, this may not be true for the tail-generated sonations. In broad-tailed hummingbirds, the minimum airspeed required for tail-generated sonations is below the speeds attained during level flight[32]. This indicates that the tail-generated sonations are not constrained to the nadir of the dives physiologically (i.e., are not fixed); instead, synchrony with visual components may be a product of selection. The dives of other species of bee hummingbirds are diverse, ranging from U-, J-, or O-shapes to undulating lines[32]. How variation in dive shape influences the synchronization of the acoustic and visual components remains to be seen. Recent work on the evolution of maneuverability in hummingbirds[46] presents new opportunities for combining the study of courtship dives with state-of-the-art methods for assessing flight skill and biomechanical performance.

Due to the difficulty of quantifying spatiotemporal variation in animal signals[1], motion and orientation are often neglected in studies of animal communication[10]. Encouragingly, this paradigm is changing[24]. A recent study by Clark et al.[51] revealed that male Costa's hummingbirds (*Calypte costae*) may have evolved a dive display that minimizes Doppler shift, potentially preventing females from using sound to extract information about a male's speed. In addition, recent work on butterflies[52,53], peacocks[3,4], and hummingbirds[38,54] suggested that males strategically orient themselves to maximize the conspicuousness of their iridescent colors, in some cases by orienting relative to the sun. Broad-tailed hummingbirds do not appear to orient dives toward the sun[37] (Hogan and Stoddard, pers. obs.; R. Simpson, pers. comm.), in contrast to Anna's hummingbird[33,35], and successive dives occur in opposite directions (Supplementary Fig. 6). Therefore, it is unlikely that males take advantage of solar position to modulate or enhance the appearance of their iridescent gorgets. However, our results show that males tend to perform dives along a trajectory that makes their gorgets both visible and dynamic (changing color rapidly) at the point of closest approach to females, potentially enhancing this visual component. Moving forward, the 'bee' hummingbirds could be a powerful model system in which to investigate multimodal and dynamic signaling. They have diverse but stereotyped courtship displays, vividly iridescent gorgets, well-understood sonations, and flight behavior that can be tracked and quantified[46,51].

## Methods

**Recording and tracking dives**. We recorded videos of broad-tailed hummingbird courtship dives in June 2017 at the Rocky Mountain Biological Laboratory in Gothic, Colorado, USA. Dives were recorded in an area near a hummingbird feeder, which was not part of a permanent male territory. A tripod and camera were positioned facing a willow shrub (*Salix* spp.), which was a frequent perch for wild female broad-tailed hummingbirds and a frequent target for visiting male broad-tailed hummingbird dives. We did not elicit dives but instead relied on the natural behavior of the males and females in the area.

Dives were filmed using a GoPro Hero 5 Black camera (GoPro, Inc., CA, USA) at 120 frames per second at a resolution of 1280:960, using the wide FOV setting (149.2° diagonal FOV), with stereo audio recorded at 48 kHz. Dives were collected between 8 am and 8 pm, and only dives with trajectories approximately orthogonal to the camera, and above a meter stick oriented vertically at 9 meters from the camera, were recorded. Visual inspection of the raw video footage led to removal of recordings that contained movements that were obviously not close to orthogonal to the camera, or represented movements other than dives. The remaining dives were digitized and their trajectories plotted. Those dives whose trajectories were atypical (in shape, indicating non-dive or non-orthogonal movement) were removed, and any remaining dives with unreasonably high maximal speed were removed (similarly indicating non-orthogonal movement). A threshold of 40 m s$^{-1}$ was chosen because (1) this is much higher than the fastest recorded flight and dive speeds in similar hummingbirds[33], and (2) it represented apparent departure from normality in a histogram of the distribution of all dives' maximal speeds. This process resulted in video clips of 48 dives in total, taken from 17 individual video recordings. It is likely that these 17 recordings capture the diving behavior of several male individuals, but because male identity was unknown, we cannot know how many. We estimated the population size of broad-tailed hummingbirds to be 250–300 at RMBL.

These videos were linearized using camera calibration parameters generated using the camera calibration toolkit in Matlab (The MathWorks, Inc., Natick, MA,

USA), created using an A4 calibration checkerboard. Mean calibration error estimated by the toolkit was 0.3 pixels at ~2 m. Diving male hummingbirds were tracked manually in these linearized videos using the digitizing tool DLTdv[55] in Matlab. The positions of birds in frames where they could not be tracked were interpolated using inpaint_nans (Matlab file exchange select, https://www.mathworks.com/matlabcentral/fileexchange/4551-inpaint-nans), and bird trajectories were smoothed using a loess filter with a span of 20% of the frames in each dive. Tracked positions were then transformed from image space to real-world space by reference to calibration images collected of a building with suitable landmarks, while replicating the viewing geometry during data collection (Supplementary Fig. 1).

Our approach to video capture and tracking is simple and accessible but relies on some estimation and post-hoc spatial calibration. One important caveat is that the judgement of whether a dive is sufficiently orthogonal to the camera to be tracked is somewhat subjective. While we believe that the general shape of the dive will be well represented in the data, our measurements of the absolute speed, velocity, and the magnitude of Doppler shift should be treated as coarse estimates. Hummingbirds have been tracked using more specialized equipment[33,46,56], but that work was designed to measure fine-scale detail such as the wing beat frequency and mechanisms of tail sonations, or highly detailed measurements of flight performance, whereas here generally only the gross features, such as the shape and height of the dive, along with the relative timing of events during the dive, were required.

**Estimating performance parameters**. Using the tracked position of males over time, we calculated a variety of measures of flight performance including speed, horizontal and vertical velocity, and the top speed achieved in each dive. These measures were calculated in an inertial reference frame where $y$ is orthogonal to gravity. Speed and velocity were calculated between frames by dividing the distance traveled (in $x$, $y$, or both) by the time interval between frames. We first averaged measures from all dives within a bout since all dives within a bout were performed by the same individual (and bouts likely represented multiple different individuals, though see above). We then averaged across bouts to obtain overall average measures of flight performance.

**Quantifying sound**. Audio was extracted from the raw footage of dives and converted into WAV format as an average of the stereo channels, and was analyzed in Raven Pro 1.5 (Bioacoustics Research Program, Ithaca, NY, USA), using Blackman-type windows with a width of 512 samples, hop size of 128 samples, and DFT size of 1024 samples. The synchronicity of the video frames to the audio recorded was corrected, and the delay in sound reaching the camera was estimated and the appropriate adjustments were made (see Supplementary Methods).

To quantify the timing of the tail-generated sonations for the dives that were tracked during the experiment, the start and end times of these sonations were manually selected on corresponding sonograms of each dive. Where this was not possible, because of noise or other interference interrupting sound recording, values were omitted, which means that we had corresponding timings for tail-generated sonations for only 35 of the 48 video-tracked dives.

**Quantifying iridescent color**. We designed a custom 3D-printed stage to photograph male broad-tailed hummingbird specimens from systematically varied angles. Photographs were taken in raw format using a calibrated, UV-sensitive Nikon D7000 camera, and a Nikkor 105 mm lens (ISO 400, aperture f/11). Each view of the specimen contained a Labsphere 40% reflectance standard on the same plane as the specimen. We captured two images, through a Baader UV/IR-Cut/L filter (420–680 nm pass) and a Baader U-Filter (320–380 nm pass), respectively. Photographs were illuminated by diffuse light from two 50 W Exo-Terra Sunray halogen lamps (Hagen Inc./Exo-Terra, Montreal, Quebec, Canada). The light was diffused by polytetrafluoroethylene (PTFE) sheeting. Between photographs, the diffuse lighting was kept constant, as were the position and angle of the camera, so that only the angle of the specimen was manipulated.

During male broad-tailed hummingbird dives, the direction of light is likely to influence the appearance of the gorget to the female. However, broad-tailed hummingbirds do not orient their dives with respect to the sun and often dive in cloudy conditions. In addition, in many cases, the very steep geometry of the dive would seem to prevent direct sunlight from falling on the gorget, while it is visible to the female, at all but very low solar elevations (for instance at sunrise or sunset). Because of this, and to generate data that would represent an average appearance of the gorget in relation to the angle of the observer, a methodology using diffuse light was chosen. Conceptually, diffuse lighting captures an approximate average of the gorget under all possible angles of lighting and should reflect common real-world conditions during dives (i.e., on overcast days, or when the gorget is not directly lit by sunlight during dives).

The resulting photographs were linearized, standardized and converted to multispectral images using the Mica toolbox[57] for ImageJ (National Institutes of Health, Bethesda, MD, USA). These images were then converted to predicted hummingbird cone stimulations in the same program, using cone sensitivities for hummingbird vision extracted from Ödeen & Håstad[39] and double cone sensitivity of the domestic chicken (*Gallus gallus domesticus*) extracted from Osorio, Vorobyev & Jones[40]. For all angles, the whole gorget was manually selected to find

mean predicted cone stimulation from light reflected off the gorget as a function of specimen angle. Photographs were taken at intervals of 20° rotation along the long axis of the bird. Values within the maximum range of angles tested were interpolated by applying Matlab's smoothing spline function to measured values. The final measures of interest were the mean LWS stimulation and the perceived luminance (double cone stimulation) of the gorget according to hummingbird vision models, as a function of the angle of the specimen. We also generated a measure of color shift, here defined as the rate of change in the LWS stimulation from the gorget. While any number of other color measures could be included, we considered these measures to be among the most apparently salient. How a female might perceive the rapid color change (see main text) is an open question, but measured flicker-fusion speeds of the Anna's hummingbird[58] suggest that in principle the female is likely to have the capacity to see and resolve the entire color change rather than just a split-second bright red flash.

We photographed ten male broad-tailed hummingbird specimens from the American Museum of Natural History (Supplementary Table 3). Additionally, one European robin (*Erithacus rubecula*) and one black-winged red bishop (*Euplectes hordeaceus*) were photographed as controls for this technique, as these birds have non-iridescent red coloration on the breast and throat, respectively. The controls did not change in hue as a function of angle, but did change in luminance in a similar fashion to hummingbird gorgets (Supplementary Fig. 8, Fig. 1). This is because while the measured patches on control birds are not iridescent, they are not perfectly diffuse—and therefore have a spectral reflectance component related to viewing geometry[59].

To further validate the use of UV/VIS photography to capture the color of the gorget as a function of specimen angle, additional tests were undertaken using a spectrophotometer and goniometer. The goniometer allows measurements of reflectance to be taken while systematically varying both the angle of measurement and the direction of lighting. The results corresponded well with the findings presented here (Supplementary Fig. 9).

**Female position and male orientation**. To examine the female's perspective and investigate the timing of different aspects of the dive, we required the female's position. This could not be measured from the recorded videos, but studies on broad-tailed hummingbird dives indicate that the receiver is <1 m below the nadir of the dive[32,35], and personal observations support this. Therefore, in this study we assumed that female position is 0.5 m below the lowest point of the dive. While this is the simplest possible assumption about female position, tests of its robustness can be found in Supplementary Note 1 (Supplementary Figures 2–4). Additionally, to understand the female's perception of the gorget, the male's posture and orientation during flight had to be assumed. The assumptions made were that the male has a similar body posture to that of the specimen (apart from wing position), i.e., that the angle of his head was coincident with that of his body, and that at all times during the dive his head and body were aligned with his movement vector. These assumptions were based on the high-speed nature of the dive: it seems likely that his body will be oriented to minimize drag. To our knowledge, changes in the posture of broad-tailed hummingbird males, apart from the spreading of tail feathers to produce tail-generated sonations, during the descent or at the nadir of the dive are not documented. However, there remains the possibility that the male orients his head or body to increase the visibility of the gorget to the female during the dive. A test of the effect that such movements can have on the timing of the visibility of the gorget during the dive can be found in Supplementary Note 2 (Supplementary Fig. 5).

These assumptions allowed us to estimate the angle at which the female is viewing the male at all points during the dive (see Supplementary Methods) and calculate the time at which the gorget becomes and then ceases to be visible. We also calculated the LWS and double cone stimulation from the gorget over time.

**Ethics**. Hummingbirds were studied under protocols approved by the RMBL Animal Use and Care Committee.

## Data availability
Data supporting this study are found in Supplementary Data 1. A reporting summary for this Article is available as a Supplementary Information file.

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

## Acknowledgements

We are grateful to D. Inouye, B. Inouye, N. Underwood, and colleagues at RMBL for logistical support, to P. Sweet and the AMNH for access to specimens, to C. Clark for valuable comments on an early draft of the manuscript, to A. Miller for assistance with data collection, to J. Weaver for assistance with 3D print design, and to colleagues at Princeton and members of the Stoddard Lab for helpful feedback. Funding provided by Princeton University and a Sloan Research Fellowship to M.C.S.

## Author contributions

M.C.S. conceived the original study, which was designed by B.H. and M.C.S. B.H. and M.C.S. collected the data and planned the analyses, which were conducted by B.H. B.H. designed the 3D-printed rotating stage. B.H. and M.C.S. wrote the paper.

## Additional information

**Competing interests:** The authors declare no competing interests.

