## [Peer Review File · Nature Communications]

Reviewers' Comments:

Reviewer #1:

Remarks to the Author:

This is an interesting and integrative study that combines analyses of hummingbird flight trajectories, sonations, and color presentation and perception. The field and analytical methods seem well executed and sophisticated. The main conclusion of the study is that the display dives are highly synchronized and stereotypical. This result will be of significant interest to all biologists working with hummingbirds, and to multiple investigators working on flight with other animals such as *Drosophila*. I believe it may also be of interest in the field of animal communication, but that topic is outside my area of expertise.

I have only one request for clarification. It was not clear how sonations were assigned as being wing-generated or tail-generated, as opposed to vocal. Was this from the video recordings? If so, was the video frame rate sufficient to assign a frequency to tail or wing movements? I do not believe these details were covered in either the methods or the supplementary materials.

I also offer one stylistic suggestion. Multiple sentences begin with the phrase "in order to". Striking "in order" from that phrase conveys the same meaning and, in my opinion, reads more easily.

Reviewer #2:

Remarks to the Author:

In this study, the authors evaluated the complex dive displays of broad-tailed hummingbirds. They found that males seem to synchronize the acoustics of the dive display with a rapid color flash at the nadir of the dive. They used various sophisticated filming, behavioral mapping, and color assessment methods to obtain their data.

Overall, I believe this study documents a very cool natural phenomenon. And methods used to obtain these measurements are clever and interesting. This work also represents a growing and important body of work on understanding how multi-modal signals are used and how they are actually perceived by receivers dynamically, instead of taking spot color and sound measures and assuming those represent the signals.

I do have some worries about the assumptions taken during the color measurements however. This study is trying to document the color presentation during a dive display in the field, but the color measurements are based off of photos taken in a museum with diffuse lighting. How males orient their dives relative to the sun would have a great effect on their coloration, which has been shown in peacocks, butterflies, and hummingbirds. Information on how males orient these dives relative to the sun is also not present. Are males consistent across individuals? Are males consistent within displays? Do males dive back and forth, or always in the same direction? I have additional qualms about the assumption of the male's orientation and position during the dives (see comments below) and would like to see these assumptions verified in some way. Finally, based on the female hummingbird flicker fusion rates, would she be able to perceive this entire change in color, or would it mostly just appear as a bright red flash to her?

Finally, the introduction and discussion were a bit short for me, especially the introduction. I would like to have more background on synchronization, why it exists, and why some species do not synchronize their displays (e.g. *Monarcha flycatchers* – Uy & Safran 2013). As of now, this reads mostly like a natural history observation, albeit a very unique and interesting observation, and I think more discussion of the bigger picture and what this study tells us about animal communication in

general is needed in both the introduction and discussion.

Additional minor comments follow below:

Introduction

Line 20: Do the authors not need to include the scientific names here?

Line 27: Add "tribe" before Mellisuginae

Line 31: I do not understand the author's use of the word "powered" here

Methods

Lines 79-82: How was a dive considered atypical? And these higher speeds that were calculated, where those due to a possible measurement error or do they represent an outstandingly fast hummingbird?

Lines 83-85: Where these dives taken all at the same place? Or on different territories? What did the authors use to elicit the displays?

Lines 212-213: How many bouts were obtained per individual?

Lines 235-238: But how does this lighting set up compare to the ambient light during dives? Do males dive in certain orientations with respect to the sun, as other hummingbird species do? Do they only dive during sunny conditions?

Lines 277-284: This is a pretty big assumption. The perception of iridescent plumage can get greatly altered even by small changes in male head orientation, and it seems reasonable that the male could change the tilt of his head to increase the amount of time his gorget is viewable. Is there no way for the authors to possibly verify this assumption? Also, how would all of this change if the males dove to the side of females instead of right above her?

We wish to thank the Editor and both reviewers for their insightful and encouraging comments on our manuscript. We have incorporated several revisions in response to your suggestions. Please find our detailed responses to comments below (in bold).

Reviewers' comments:

Reviewer #1 (Remarks to the Author):

This is an interesting and integrative study that combines analyses of hummingbird flight trajectories, sonations, and color presentation and perception. The field and analytical methods seem well executed and sophisticated. The main conclusion of the study is that the display dives are highly synchronized and stereotypical. This result will be of significant interest to all biologists working with hummingbirds, and to multiple investigators working on flight with other animals such as *Drosophila*. I believe it may also be of interest in the field of animal communication, but that topic is outside my area of expertise.

We appreciate this – thank you!

I have only one request for clarification. It was not clear how sonations were assigned as being wing-generated or tail-generated, as opposed to vocal. Was this from the video recordings? If so, was the video frame rate sufficient to assign a frequency to tail or wing movements? I do not believe these details were covered in either the methods or the supplementary materials.

Thank you very much for pointing this out. We assigned sounds as either wing- or tail- generated based on detailed descriptions by Clark et al. 2012. In that paper they describe the dive sounds and wing- and tail-feather morphology of broad-tailed hummingbirds, black-chinned hummingbirds, and a broad-tail X black-chinned hybrid. In manipulations where a broad-tail's tail feathers were plucked, the bird could no longer generate the sounds assigned as tail-generated in the main text. In addition, sounds similar to those assigned as tail- or wing- generated in the main text can be produced by putting tail and wing feathers, respectively, in wind tunnels. Additional work by Miller and Inouye 1983 experimentally silenced the wing-generated noises of broad-tailed hummingbirds by gluing wing primaries together.

We now clarify this point L96-99:

“Audio analysis revealed that the wing- and tail-generated sonations of broad-tailed hummingbird courtship dives are highly stereotyped and can be separated into three general sections that always occurred in a precise order (Fig. 1a & b, Supplementary Fig. S7). The sonations were assigned as wing- or tail- generated based on previous descriptions of these sounds 35,36”

I also offer one stylistic suggestion. Multiple sentences begin with the phrase “in order to”. Striking “in order” from that phrase conveys the same meaning and, in my opinion, reads more easily.

Thank you for this suggestion – we agree. We now omit all instance of ‘in order’ – e.g., lines 289, 293, 339, 345.

Reviewer #2 (Remarks to the Author):

In this study, the authors evaluated the complex dive displays of broad-tailed hummingbirds. They found that males seem to synchronize the acoustics of the dive display with a rapid color flash at the nadir of the dive. They used various sophisticated filming, behavioral mapping, and color assessment methods to

obtain their data.

Overall, I believe this study documents a very cool natural phenomenon. And methods used to obtain these measurements are clever and interesting. This work also represents a growing and important body of work on understanding how multi-modal signals are used and how they are actually perceived by receivers dynamically, instead of taking spot color and sound measures and assuming those represent the signals.

Thank you!

I do have some worries about the assumptions taken during the color measurements however. This study is trying to document the color presentation during a dive display in the field, but the color measurements are based off of photos taken in a museum with diffuse lighting.

We now provide more details on the diffuse lighting and our rationale for using museum specimens (we have consolidated this discussion with a similar point, see below.)

How males orient their dives relative to the sun would have a great effect on their coloration, which has been shown in peacocks, butterflies, and hummingbirds. Information on how males orient these dives relative to the sun is also not present. Are males consistent across individuals? Are males consistent within displays? Do males dive back and forth, or always in the same direction?

Thank you for this thoughtful comment. It's true that orientation relative to the sun would affect the appearance of the gorget in any given dive. However, male broad-tailed hummingbirds do not appear to orient their dives toward (or relative to) the sun (Hogan and Stoddard pers. obs.; R. Simpson pers. comm.). Further evidence comes in Hamilton's (1965) assertion that "The Anna's Hummingbird ... is, as far as is known, the only species of hummingbird which orients its display with respect to the sun". Broad-tailed hummingbirds also do not orient toward the sun (Simpson & McGraw, 2018) during another display type – the 'shuttle' display. Males also undertake dives in alternating directions along more or less the same trajectory – see figure S6 for an idea of the symmetry within bouts of dives. We've also observed them dive during sunny and cloudy weather conditions, and at all times of day. Unfortunately we do not have information on whether individual males are consistent in orientation between bouts, though we have occasionally observed changes in orientation within a bout (in response to female movement), so we suspect that males are not constrained to dive in any one (or two) directions but rather adjust their dive trajectories based on the female's position.

We have clarified the main text with the following.

Lines 211-220 now read:

"In addition, recent work on butterflies 52,53, peacocks 3,4, and hummingbirds 38,54 suggested that males strategically orient themselves to maximize the conspicuousness of their iridescent colors, in some cases by orienting relative to the sun. Broad-tailed hummingbirds do not appear to orient toward the sun (Hogan and Stoddard, pers. obs.; R. Simpson, pers. comm.; 37, in contrast to Anna's hummingbird; 33,35), and successive dives occur in opposite directions (see supplemental Fig. S6). Therefore, it is unlikely that males take advantage of solar position to modulate or enhance the appearance of their iridescent gorgets. However, our results show that males tend to perform dives along a trajectory that makes their gorgets both visible and dynamic (changing color rapidly) at the point of closest approach to females, potentially enhancing this visual component."

I have additional qualms about the assumption of the male's orientation and position during the dives (see comments below) and would like to see these assumptions verified in some way.

Excellent point, our response can be found below (consolidated with a similar point for clarity).

Finally, based on the female hummingbird flicker fusion rates, would she be able to perceive this entire change in color, or would it mostly just appear as a bright red flash to her?

This is an interesting question. Evidence from the closely-related Anna's hummingbird indicate that hummingbird flicker fusion frequencies are between 70 and 80 Hz. If we take the standpoint that eyes work like cameras, the female might see 9 'frames' while the gorget is visible and changing color ($0.12s/(1s/75Hz)$). Whether or not this appears simply as a red-to-black blur, or as a bright red flash is hard to predict. However, in principle a female has the capacity to see and resolve the entire color change, rather than just a split-second bright red flash.

1. Fellows T. (2015) Visual resolution of Anna's hummingbirds (*Calypte anna*) in space and time. MSc thesis (University of British Columbia, Vancouver, Canada). Available at <https://open.library.ubc.ca/cIRcle/collections/ubctheses/24/items/1.0166301>

We have included the following to the main text which may allow interested readers to draw their own conclusions, at L325-328:

"How a female might perceive the rapid color change found (see main text) is an open question, but measured flicker-fusion speeds of the Anna's hummingbird 58 suggest that in principle the female has the capacity to see and resolve the entire color change rather than just a split-second bright red flash."

Finally, the introduction and discussion were a bit short for me, especially the introduction. I would like to have more background on synchronization, why it exists, and why some species do not synchronize their displays (e.g. Monarcha flycatchers – Uy & Safran 2013). As of now, this reads mostly like a natural history observation, albeit a very unique and interesting observation, and I think more discussion of the bigger picture and what this study tells us about animal communication in general is needed in both the introduction and discussion.

We have taken this suggestion seriously and have added several paragraphs to the introduction and discussion about the presumed functions of multimodal signaling and synchronization. We now place our results in a more developed conceptual framework.

See lines 32-63 & 175-191.

Additional minor comments follow below:

Introduction

Line 20: Do the authors not need to include the scientific names here?

Thank you – we have added this. Lines 24-26 now read:

"Not only are these displays often multimodal, stimulating multiple senses, they typically include motion: peacocks (*Pavo cristatus*) rattle their trains 3,4, jacky dragons (*Amphibolurus muricatus*) flick

their tails 5, and spiders (*Habronattus spp.*, *Schizocosa spp.*) dance, producing vibratory signals in the process 6–9”

Line 27: Add “tribe” before Mellisuginae

Line 65 now reads:

“The ‘bee’ hummingbirds (tribe Mellisuginae), a colorful and rapidly diversifying clade 31, provide a powerful system in which to investigate complex, dynamic displays.”

Line 31: I do not understand the author’s use of the word “powered” here.

Thank you for pointing this out. We used “powered” to convey that the bird actively flaps its wings to propel itself during (at least most) of the dive. We have clarified in L68-70:

“Males display to stationary females by climbing vertically to approximately 30m and making sequential powered (by active wing-flapping) U-shaped dives 32,35.”

Methods

Lines 79-82: How was a dive considered atypical? And these higher speeds that were calculated, where those due to a possible measurement error or do they represent an outstandingly fast hummingbird?

We suspect dives with unrealistically high speeds are the results of measurement error. More specifically, we suspect that dives containing such speeds must contain movements that are not orthogonal to the camera (thus distorting our trajectory). We have clarified both points in L239-245:

“The remaining dives were digitized and their trajectories plotted. Those dives whose trajectories were atypical (in shape, indicating non-dive or non-orthogonal movement) were removed, and any remaining dives with unreasonably high maximal movement speed were removed (similarly indicating non-orthogonal movement). A threshold of 40m/s-1 was chosen because 1) this is much higher than the fastest recorded flight and dive speeds in similar hummingbirds 33, and 2) it represented apparent departure from normality in a histogram of the distribution of all dives’ maximal speeds.”

Lines 83-85: Where these dives taken all at the same place? Or on different territories? What did the authors use to elicit the displays?

We have clarified the methodology used L227-232:

“We recorded videos of broad-tailed hummingbird courtship dives in June 2017 at the Rocky Mountain Biological Laboratory in Gothic, Colorado, USA. Dives were recorded in an area near a hummingbird feeder, which was not part of a permanent male territory. A tripod and camera were positioned facing a willow shrub (*Salix spp.*), which was a frequent perch for wild female broad-tailed hummingbirds, and a frequent target for visiting male broad-tailed hummingbird dives. We did not elicit dives but instead relied on the natural behavior of the males and females in the area.”

Lines 212-213: How many bouts were obtained per individual?

As noted in lines 245-248, we are unable to ascertain the individual identity of males performing dives. Our estimates of the population size (L248: 250-300) and personal observations support the

assertion that male identity likely differed between bouts, but we do not know how many unique individuals we observed. We have noted this in L275-277:

“We first averaged measures from all dives within a bout since all dives within a bout were performed by the same individual (and bouts likely represented multiple different individuals, though see above). We then averaged across bouts to obtain overall average measures of flight performance.”

See also lines 245-248:

“This process resulted in video clips of 48 dives in total, taken from 17 individual video recordings. It is likely that these 17 recordings capture the diving behavior of several male individuals, but because male identity was unknown, we cannot know how many. We estimate the population size of broad-tailed hummingbirds to be 250-300 at RMBL.”

Lines 235-238: But how does this lighting set up compare to the ambient light during dives? Do males dive in certain orientations with respect to the sun, as other hummingbird species do? Do they only dive during sunny conditions?

This is an important point, thank you for prompting us to clarify. We choose to photograph the gorgets under diffuse light for a number of reasons:

- 1. Broad-tailed hummingbirds do not appear to orient dives with respect to the sun, and will dive in overcast conditions.**
- 2. In the main text, we aimed to obtain a general measure of gorget coloration that depended on the angle/direction of the signal observer rather than on the angle of direct sunlight. Overall, such a measure would represent the average appearance of the gorget, irrespective of the location of the sun. Using diffuse light allowed us to achieve this: diffuse light is conceptually similar to an integrated appearance of the gorget across all possible angles of directional lighting. Additionally, diffuse light provides lighting conditions similar to those of a cloudy day, with more or less uniformly scattered light from the background (we think this is a common lighting condition for the display – see point 1&3). See Osorio and Ham “Spectral reflectance and directional properties of structural coloration in bird plumage” for an example that uses diffuse light to quantifying bird plumage color.**
- 3. Given the geometry of the BTH’s dive, there seem to be few solar elevations at which the gorget of the male will be lit directly by the sun while it is visible to the female during dives (see illustration below). This is because while the gorget is visible to the female in a dive, the male is nearly horizontal (beak pointing to the horizon). This indicates to us that only when the sun is low in elevation (for instance at sunrise or sunset) is the gorget likely directly lit by the sun. At all other solar positions it seems likely that the body of the male obstructs direct lighting from the sun, so that the gorget will be lit by scattered (more or less diffuse) light from the background. Interestingly, this may also be true to an extent for the dives of Anna’s hummingbird which *are* sun-oriented. However, there the birds also have an iridescent crown (which likely makes color clearly visible throughout most of the dive; https://www.youtube.com/watch?v=ayzg_5YEQ7w), and the trajectory of the dive is shallower.**

Figure 1 - Illustration of point 3. Colored sections indicate range of solar elevations (if a male was oriented toward the sun) when direct sunlight might be able to fall on the gorget while it is visible to the female.

We agree that these points deserve clarification in the main text:

Lines 211-220 (discussion) now read:

“In addition, recent work on butterflies 52,53, peacocks 3,4, and hummingbirds 38,54 suggested that males strategically orient themselves to maximize the conspicuousness of their iridescent colors, in some cases by orienting relative to the sun. Broad-tailed hummingbirds do not appear to orient toward the sun (Hogan and Stoddard, pers. obs.; R. Simpson, pers. comm.; 37, in contrast to Anna’s hummingbird; 33,35), and successive dives occur in opposite directions (see supplemental figure S6). Therefore, it is unlikely that males take advantage of solar position to modulate or enhance the appearance of their iridescent gorgets. However, we find that the male may be strategically choosing a dive trajectory that makes his gorget both visible and dynamic (changing color rapidly) at the point of closest approach to the female, potentially enhancing this visual component.”

Lines 303-311 (methods) now read:

“During male broad-tailed hummingbird dives, the direction of lighting is likely to influence the appearance of the gorget to the female. However, broad-tailed hummingbirds do not orient their dives with respect to the sun, dive in cloudy conditions, and in many cases the very steep geometry of the dive would seem to preclude direct sunlight falling on the gorget while it is visible to the female at all but very low solar elevations. Because of this, and to generate data that will represent an average appearance of the gorget in relation to the angle of the observer, a methodology using diffuse light was chosen. Conceptually, diffuse lighting captures an approximate average of the gorget under all possible angles of lighting, and should reflect common actual lighting conditions during dives (i.e. on overcast days, or when the gorget is not directly lit by sunlight during dives).”

Lines 277-284: This is a pretty big assumption. The perception of iridescent plumage can get greatly altered even by small changes in male head orientation, and it seems reasonable that the male could change the tilt of his head to increase the amount of time his gorget is viewable. Is there no way for the authors to possibly verify this assumption?

It certainly does seem reasonable that the male might change head position, and indeed we did some work to verify this assumption in the supplemental material (see lines 355-360):

“To our knowledge, changes in the posture of broad-tailed hummingbird males, apart from the spreading of tail feathers to produce tail-generated sonations, during the descent or at the nadir of the dive are not documented. However, there remains the possibility that the male orients his head or body to increase the visibility of the gorget to the female during the dive. A test of the effect that such movements could have had on the timing of the visibility of the gorget during the dive can be found in the supplementary information (Supplementary Fig. S5).”

In this supplemental analysis we artificially increased the angles over which the gorget is visible to the female, by 20, 40, and 60 degrees. We found that this does increase the amount of time the gorget is visible during the dive. However, even in the latter most extreme case, the portion of time that the gorget is visible still coincides with the time that male is closest to the female, that is – the results are qualitatively unchanged (and color can only be relevant to the dive while the gorget is visible to the female). Our interpretation is that the timing of this part of the dive is defined more by the U-shaped geometry of the dive than the exact angles over which the gorget is visible.

Also, how would all of this change if the males dove to the side of females instead of right above her?

All hummingbirds apart from Costa’s are thought to dive above and not next to the female (Clark and Mistick, 2018; and our observations place the broad-tailed males over the female). It’s an interesting point though. If he were to the side of her then the relative timing of events during the dive are unlikely to change, apart from (potentially) the actual appearance of the gorget, and the absolute magnitude of the Doppler effect during the dive (see Clark and Mistick 2018 for very interesting discussion on what this influence on the Doppler effect might mean for information transfer to the female).

If instead he dives such that he reaches the nadir before or after passing over the female, there are changes to the synchronization of the parts of the dive. We included a simulation of such a situation in the supplemental material (section ‘Female position’). We found that the male may actually be able to slightly increase the synchronization between the tail sonations and visibility of the gorget if he passes over the female before he reaches the nadir of the dive. Thus, his ability to accurately target the female could be a very important factor in the synchronization of the display components, which could potentially be correlated with his quality (flying or foraging ability). Exploring this in the future – and really getting at inter-individual variation in the dives – would be very exciting. For example, we might generate the prediction that on average we should see males passing over the female before the nadir if synchronization really is important to the female.

Reviewers' Comments:

Reviewer #1:

Remarks to the Author:

The authors have fully addressed my one minor concern. I also like the changes made to the manuscript in response to the other reviewer's concerns. I am happy to endorse publication.

Reviewer #2:

Remarks to the Author:

The authors did an excellent job addressing my comments and I also commend them for their fleshed-out conceptual framework.

I only have one minor suggestion, if it is possible within the word limit and/or flow of the manuscript: Would it be possible to incorporate your comment that only when the sun is at a low elevation (for instance at sunrise or sunset) is the gorget likely directly lit by the sun in the manuscript? Perhaps somewhere in lines 211-220 or 303-311? I ask because, to me, this wording really drives home the point that the sun is unlikely to be of any consequence to these male broad-tailed hummingbirds as they dive.

Response to reviewers

Please see responses in bold.

Reviewer #1 (Remarks to the Author):

The authors have fully addressed my one minor concern. I also like the changes made to the manuscript in response to the other reviewer's concerns. I am happy to endorse publication.

Thank you!

Reviewer #2 (Remarks to the Author):

The authors did an excellent job addressing my comments and I also commend them for their fleshed-out conceptual framework.

Thank you!

I only have one minor suggestion, if it is possible within the word limit and/or flow of the manuscript: Would it be possible to incorporate your comment that only when the sun is at a low elevation (for instance at sunrise or sunset) is the gorget likely directly lit by the sun in the manuscript? Perhaps somewhere in lines 211-220 or 303-311? I ask because, to me, this wording really drives home the point that the sun is unlikely to be of any consequence to these male broad-tailed hummingbirds as they dive.

The lines (previously at 303-311) now read;

“During male broad-tailed hummingbird dives, the direction of light is likely to influence the appearance of the gorget to the female. However, broad-tailed hummingbirds do not orient their dives with respect to the sun and often dive in cloudy conditions. In addition, in many cases, the very steep geometry of the dive would seem to prevent direct sunlight from falling on the gorget, while it is visible to the female, at all but very low solar elevations (for instance at sunrise or sunset).”